

# Fossil eggshell cuticle elucidates dinosaur nesting ecology

Tzu-Ruei Yang[1], Ying-Hsuan Chen[2], Jasmina Wiemann[3], Beate Spiering[4] and P. Martin Sander[1,5]

[1] Bereich Paläontologie, Steinmann-Institut für Geologie, Mineralogie und Paläontologie, Universität Bonn, Bonn, Germany
[2] Max-Planck-Institut für Eisenforschung, Düsseldorf, Germany
[3] Department of Geology and Geophysics, Yale University, New Haven, CT, United States of America
[4] Bereich Mineralogie, Steinmann-Institut für Geologie, Mineralogie und Paläontologie, Universität Bonn, Bonn, Germany
[5] Dinosaur Institute, Natural History Museum of Los Angeles County, Los Angeles, CA, United States of America

Corresponding author
Tzu-Ruei Yang, tryang@uni-bonn.de, lereage@gmail.com

## ABSTRACT

The cuticle layer consisting mainly of lipids and hydroxyapatite (HAp) atop the mineralized avian eggshell is a protective structure that prevents the egg from dehydration and microbial invasions. Previous ornithological studies have revealed that the cuticle layer is also involved in modulating the reflectance of eggshells in addition to pigments (protoporphyrin and biliverdin). Thus, the cuticle layer represents a crucial trait that delivers ecological signals. While present in most modern birds, direct evidence for cuticle preservation in stem birds and non-avian dinosaurs is yet missing. Here we present the first direct and chemical evidence for the preservation of the cuticle layer on dinosaur eggshells. We analyze several theropod eggshells from various localities, including oviraptorid *Macroolithus yaotunensis* eggshells from the Late Cretaceous deposits of Henan, Jiangxi, and Guangdong in China and alvarezsaurid *Triprismatoolithus* eggshell from the Two Medicine Formation of Montana, United States, with the scanning electron microscope (SEM), electron probe micro-analysis (EPMA), and Raman spectroscopy (RS). The elemental analysis with EPMA shows high concentration of phosphorus at the boundary between the eggshell and sediment, representing the hydroxyapatitic cuticle layer (HAp). Depletion of phosphorus in sediment excludes the allochthonous origin of the phosphorus in these eggshells. The chemometric analysis of Raman spectra collected from fossil and extant eggs provides further supportive evidence for the cuticle preservation in oviraptorid and probable alvarezsaurid eggshells. In accordance with our previous discovery of pigments preserved in Cretaceous oviraptorid dinosaur eggshells, we validate the cuticle preservation on dinosaur eggshells through deep time and offer a yet unexplored resource for chemical studies targeting the evolution of dinosaur nesting ecology. Our study also suggests that the cuticle structure can be traced far back to maniraptoran dinosaurs and enhance their reproductive success in a warm and mesic habitat such as Montana and southern China during the Late Cretaceous.

## INTRODUCTION

### Cuticle structures and functions

An avian egg is an evolutionary invention that protects the developing embryo against mechanical damage, dehydration, and microbial invasion (*Romanoff & Romanoff, 1949*; *Board & Fuller, 1974*). This multi-functionality is contributed to by both mineralized and non-mineralized organic layers, which exhibit unique biomaterial properties. The avian eggshell forms inside the uterus. The final layer to be deposited, the eggshell cuticle, represents the interface between the embryo inside and its outside environment. Many bird eggs are equipped with a cuticle layer; however, this structure is absent in some clades such as parrots, petrels, and pigeons (*D'Alba et al., 2016*). The cuticle layer is often described as "waxy" due to its high amount of lipids, which keep the internal fluids from evaporating and therefore protect the encased embryo from desiccation. In addition to lipids, the avian cuticle contains proteins, polysaccharides, calcium carbonate (vaterite), calcium phosphates (hydroxyapatite, HAp), and pigments (*Wedral, Vadehra & Baker, 1974*; *Nys et al., 1991*; *Packard & DeMarco, 1991*; *Dennis et al., 1996*; *Mikhailov & British Ornithologists' Club, 1997*; *Fraser, Bain & Solomon, 1999*; *Cusack, Fraser & Stachel, 2003*; *Igic et al., 2015*).

The cuticle layer varies in thickness (up to 12 µm) and has a patchy distribution on the eggshell (*Board & Halls, 1973*). Basically, the cuticle is divided into two distinct layers, including an amorphous HAp/vaterite inner layer and a proteinaceous outer layer (*Simons, 1971*; *Dennis et al., 1996*; *Fraser, Bain & Solomon, 1999*). The inner cuticle layer is composed of needle-like HAp/vaterite crystals on some avian eggs but present in the form of nanospherical HAp/vaterite on other avian eggs (*Dennis et al., 1996*; *D'Alba et al., 2014*). A recent study suggested that the circular mineral nanospheres on eggshells serve an antimicrobial function for species nesting in humid environment (*D'Alba et al., 2016*). *D'Alba et al. (2016)* also proposed that eggs in humid environments tend to have more cuticular nanospheres to prevent flooding and microbial invasions. Conversely, in arid environments, lacking a cuticle layer would be beneficial for embryos in eggs with low conductance, as *Deeming (1987)* suggested. However, in an extreme environment such as the Antarctic, Adélie penguin eggs have a thick cuticle layer to reduce water loss (*Thompson & Goldie, 1990*). Besides, the intactness of the cuticle is an indicator for the freshness of an egg (*Rodríguez-Navarro et al., 2013*). Several studies suggested that the cuticle functions as a lubricant that facilitates egg rotation in the uterus (*Rahman et al., 2009*). Moreover, a recent study shows that the cuticle also modulates ultraviolet reflectance of avian eggshells (*Fecheyr-Lippens et al., 2015*). For instance, an extremely smooth cuticle produces glossiness and iridescence in tinamou eggs (*Igic et al., 2015*).

The cuticle is commonly observed on most calcified eggshells, but it shows different structures and characteristics between reptilian and avian eggshells (*Ferguson, 1982*). The cuticle on reptilian eggshells is no longer present after two weeks of laying (*Ferguson, 1982*); hence, the author concluded that the reptilian cuticle is not equivalent to the cuticle present on avian eggshells. *Ferguson (1982)* hypothesized that a true cuticle layer is not present on alligator eggshell. Conversely, avian eggshells exhibit an inner cuticle layer composed of HAp/vaterite granules or amorphous needle-like HAp/vaterite crystals, which

are absent on reptilian eggshells. Several studies on extant eggshell have identified cuticle presence by means of the scanning electron microscopy (SEM) and transmission electron microscopy (TEM) (*Fink et al., 1993*; *Fraser, Bain & Solomon, 1999*). *Kusada et al. (2011)* used energy dispersive X-ray analysis (EDX) to map the phosphorus (P) and calcium (Ca) distribution in several extant avian eggshells and discovered that P is confined to the cuticle layer. Although the P concentration in the cuticle varies greatly between different bird species, P is a distinct marker for identifying the cuticle layer in case of P depletion in sediment. Although it is still not clear how P functions during the eggshell formation, a study suggested that the inclusion of P in the biomineral terminates eggshell growth since it competes with calcium during the eggshell crystallization (*Cusack, Fraser & Stachel, 2003*).

## Discoveries of fossilized organic membranes

Since the cuticle is composed of proteins, polysaccharides, lipids, calcium carbonate, calcium phosphates, and pigments, it is seemingly unlikely to preserve during the fossilization process. A previous report concerning spherical (*Oölithus spheroides*) and elongated (*Oölithus elongatus*) eggs from Laiyang, Shandong, China, suggested that the cuticle is preserved in elongated eggs but not in spherical ones (*Chow, 1954*). Another similar report on ornithoid eggshells from Naran-Bulak, Mongolia also described possible cuticle preservations (*Sochava, 1971*), but no diagnostic chemical data were obtained. *Kohring & Hirsch (1996)* studied the crocodilian and ornithoid eggshells from the Middle Eocene of the Geiseltal, Germany, and suggested possible cuticle and shell membrane preservation. Later, with the SEM, several studies on dinosaur eggshells, Cretaceous bird eggshells, and fossil moa eggshells showed the potential of cuticle preservation (*Mikhailov, 1991*; *Varricchio & Jackson, 2004*; *Jackson & Varricchio, 2010*; *Oskam et al., 2010*). Although these studies provided a new avenue to understanding soft-part preservation, all previous studies described possible cuticle preservation only based on micro-structural comparisons with the cuticle of extant avian eggshells. *Jackson & Varricchio (2010)* described a new ootaxon—*Triprismatoolithus stephensi*—of probable alvarezsaurid affinities. These authors reported a cracked and amorphous surface overlying the external layer and suggested that it might be a layer of cuticle. Nevertheless, they did not further investigate the possible cuticle layer by means of any other analyses. Therefore, all previous reports regarding the cuticle preservation were constrained to micro-morphological observations. The only chemical evidence supporting the fossilization potential of cuticle through deep time is offered by a study on oviraptorid eggshells focusing on the two main eggshell pigments, biliverdin and protoporphyrin (*Wiemann et al., 2017*). The protoporphyrin is deposited predominantly in the cuticle layer (*Schwartz et al., 1975*; *Nys et al., 1991*; *Mikšík et al., 2007*; *Nys & Guyot, 2011*), thus providing indirect evidence of cuticle preservation in the preservation of endogenous protoporphyrin (*Wiemann et al., 2017*). Although *Wilson et al. (2017)* demonstrated that there is no codependence of the deposition of pigment and cuticle, the preserved pigments in the oviraptorid eggshell could derive from the eggshell or the preserved cuticle layer and in the latter instance would be indicative of cuticle preservation. Hence, we posited that the cuticle was completely, or partially, preserved on the oviraptorid eggshells. In addition to the cuticle layer, the *membrana testacea* (eggshell

membranes), a proteinaceous meshwork that underlies the mineralized eggshell and separates egg yolk and eggshell, is also a crucial membrane. Unlike the cuticle layer, the preservation of *membrana testacea* has been reported from titanosaurid and *Lufengosaurus* eggs (*Grellet-Tinner, 2005*; *Reisz et al., 2013*), yet never in theropod eggs.

## Nesting ecology of dinosaurs and modern birds

Dinosaur nesting ecology is a long-standing and intriguing research topic. Among all dinosaurs, the nesting type of titanosaur, troodontid, and oviraptorid dinosaurs are comparatively well documented (*Jackson, 2007*; *Varricchio et al., 2013*; *Wiemann et al., 2017*). These dinosaurs adopted different nesting strategies in their clutch arrangement, nest architecture, and nesting mode. *Deeming (2006)* and *Tanaka, Zelenitsky & Therrien (2015)* roughly categorized two nesting modes in dinosaurs, including open and buried nesting modes. Most birds build open nests, while the buried nesting mode is only observed in megapode birds. In a wet incubation environment such as a buried megapode nest, eggs are exposed to a higher risk of asphyxiation and microbial infection (*Board & Fuller, 1974*; *Board, 1982*; *D'Alba et al., 2017*). *D'Alba et al. (2016)* provided the first report regarding the association between the cuticular nanospheres and nesting modes. These authors discovered that the birds nesting in more mesic environments tend to lay eggs yielding cuticular nanospheres. The nanospheres in the cuticle layer, therefore, represents a structural adaptation that prevents eggs from microbial invasions (*D'Alba et al., 2016*). However, eggs of Adélie penguins are also encapsulated by a thick cuticle layer that prevents eggs from water loss in the severe cold Antarctic environment (*Thompson & Goldie, 1990*). These lines of evidence suggest that the cuticle layer is an evolutionarily labile structure and thus varies greatly in accordance with the nesting environment. *D'Alba et al. (2016)* further suggested that cuticular nanospheres are an ancestral trait; however, this trait has been lost multiple times and is absent in some avian clades.

In dinosaurs, it was suggested that titanosaurs also employed the buried nesting mode (*Sander et al., 1998*; *Grellet-Tinner & Fiorelli, 2010*; *Sander et al., 2008*; *Vila et al., 2010*; *Hechenleitner, Grellet-Tinner & Fiorelli, 2015*). The low porosity of titanosaur eggs from Auca Mahuevo, however, suggested that the titanosaurs from Auca Mahuevo did not bury their eggs (*Jackson, 2007*; *Sander et al., 2008*). The Auca Mahuevo exception demonstrates that the eggshell is a labile structure, which alters according to the environment. In comparison with the well-documented titanosaur nesting ecology, the nesting ecology of theropods has remained uncertain until recent times. A nesting *Nemegtomaia* discovered in Mongolia indicated a semi-open nesting mode based on sedimentological evidence (*Fanti, Currie & Demchig, 2012*). The discovery of pigmentation in oviraptorid eggshells strongly suggested that the oviraptorid eggs were laid in an at least partially open nest (*Wiemann et al., 2017*). *Yang et al. (2015)* (see also *Wiemann et al., 2017*) further supported the semi-open nesting mode according to the heterogeneous distribution of porosity in an oviraptorid egg and taphonomic evidence. While there is no evidence of pigmentation in troodontid eggshells because these have not been analyzed chemically so far, the heterogeneous distribution of porosity suggests that the troodontid dinosaurs might also have laid their eggs in a semi-open nest (*Varricchio et al., 1997*; *Varricchio et al., 2013*).

**Table 1** **Eggshells analyzed in this study.** The ootaxonomic/taxonomic assignments, stratigraphic and locality information, and catalog numbers of the eggshells analyzed in this study.

| Oospecies or species | Stratigraphy | Locality | Catalog number | Reference |
|---|---|---|---|---|
| *Macroolithus yaotunensis* | Tangbian Formation | Hongcheng Basin, Jiangxi Province, China | NMNS CYN-2004-DINO-05-I | *Wiemann et al. (2017)* |
| *Macroolithus yaotunensis* | Hugang Formation | Liquanqiao Basin, Henan Province, China | STIPB-E131 | *Wiemann et al. (2017)* |
| *Macroolithus yaotunensis* | Pingling Formation | Nanxiong Basin, Guangdong Province, China | STIPB-E66 | *Wiemann et al. (2017)* |
| *Triprismatoolithus stephensi* | Lower portion of the Upper Cretaceous (Campanian) Two Medicine Formation | Dave and Joel Site, Sevenmile Hill outcrops, Teton County, Montana | MOR ES101 | *Jackson & Varricchio (2010)* |
| *Gallus gallus domesticus* | – | Bonn, Germany | – | – |
| *Crocodylus porosus* | – | Taichung, Taiwan | uncatalogued specimen in NMNS | – |
| *Tomistoma schlegelii* | – | Taichung, Taiwan | uncatalogued specimen in NMNS | – |

The goal of this study is to present further evidence of cuticle preservation on dinosaur eggshells by means of SEM imaging, elemental analysis, and RS. In particular, we analyzed theropod eggshells from the Nanxiong Group in China and the Two Medicine Formation in Montana, USA. Both the Nanxiong Group and Two Medicine Formation represent fluvial deposits, indicating a probable humid climate in both areas during the Late Cretaceous. In addition, nesting on the ground or in a mound in a wet environment increases the risk of microbial invasions (*D'Alba et al., 2016*), a cuticle layer should thus be prevalent on eggshells of ground nesting theropods in a humid climate.

## MATERIAL AND METHODS

### Materials

In order to standardize our following analyses, domestic chicken eggshell (*Gallus gallus domesticus*) was obtained from a commercial source (supermarket). Two extant crocodilian eggshells (*Crocodylus porosus* and *Tomistoma schlegelii*) were sampled from the collection of the NMNS as negative controls (Table 1). The fossil materials include oviraptorid and probable alvarezsaurid dinosaur eggshells (Table 1).

We studied three oviraptorid eggshell specimens that were collected from the Late Cretaceous Nanxiong Group of three localities in China (Henan Province, Jiangxi Province, and Guangdong Province) and are housed at the NMNS and STIPB. Based on eggshell microstructure, all specimens were assignable to *Macroolithus yaotunensis*, which was laid by the oviraptorid *Heyuannia huangi* (*Cheng et al., 2008*). *Macroolithus yaotunensis* is characterized by an undulating boundary between two crystalline layers, i.e., the outer prismatic layer and the inner mammillary layer (Supplemental Information;

*Zhao, 1975*). The first *Macroolithus yaotunensis* eggshell was taken from specimen NMNS CYN-2004-DINO-05-I, which was collected from the Late Cretaceous Tangbian Formation of the Hongcheng Basin in Jiangxi Province, China (Table 1). The Tangbian Formation is characterized by fine-grained brownish-reddish sediments with dispersed coarse clasts and conglomerates, representing a fluvial/alluvial environment with occasional stream flooding during the Campanian stage (*Chen et al., 2017*). The second *Macroolithus yaotunensis* sample was collected as isolated shell fragments from the Late Cretaceous Hugang Formation in the Liguanqiao Basin near Nanyang, southwestern Henan Province, China (Table 1). This sample has been housed in STIPB since 1985 and was described in *Erben (1995)*. The Hugang Formation is composed of dark grayish red breccia with interbeds of calcareous sandstone, indicating a fluvial/alluvial environment (*Zhang, 2009*). The third *Macroolithus yaotunensis* sample was also obtained as isolated shell fragments from the Late Cretaceous fluvial deposits of the Pingling Section of the Shanghu Formation in the Nanxiong Basin, located in the northwestern part of Guangdong Province, China (Table 1). The NE-SW striking Nanxiong Basin, which is filled with dark purplish silty mudstone with interbeds of conglomerate, has produced numerous theropod egg clutches and eggshell fragments (*Erben, 1995*).

The *Triprismatoolithus stephensi* eggshells (MOR ES101), probably attributable to alvarezsaurids (*Agnolin et al., 2012*; *Varricchio & Jackson, 2016*), were collected as isolated shell fragments in the lower portion of the Two Medicine Formation in the Sevenmile Hill outcrops, Teton County, Montana, USA (Table 1). The *Triprismatoolithus stephensi* eggshell has three crystalline layers, including an external layer, prismatic layer, and mammillary layer, from outermost to innermost (*Varricchio & Jackson, 2016*). The Two Medicine Formation is mostly composed of sandstone, deposited by rivers and deltas on the western shoreline of the Late Cretaceous Interior Seaway.

## Chemical analyses

For the chemical analyses, we first used a VEGA TS5130 LM (Tescan) SEM at the STIPB to locate and image plausible cuticle preservation on the studied eggshells. The eggshells were naturally broken fragments and coated with gold for SEM observation. The operating conditions for the SEM were set as 20 kV accelerating voltage.

All samples were cleaned with ethanol before chemical analyses. We then performed elemental analyses using the EPMA equipped with a WDS at the Section of Mineralogy of the STIPB to elucidate the P distribution in the radial section of dinosaur eggshells. All samples were polished, fixed on a glass slide with araldite histological resin, and then coated with carbon for the elemental analysis with EPMA. The elemental analyses were performed with a JEOL SUPERPROBE 8900 EPMA at the same lab. The operating conditions for the EPMA were set as 1 $\mu$m beam diameter, 15 kV accelerating voltage, and 15 nA specimen current. The ZAF correction scheme was used. Natural and synthetic minerals were used as standards.

We performed two rounds of RS using two Raman spectrometers at the MPIE. For the first round, Raman Analysis I (referred hereafter as RI), we analyzed the domestic chicken eggshell from a commercial source with a WITec Alpha300 system with 532 nm wavelength

laser light and a confocal 50× objective, each for 20 s. Since the cuticle on the chicken eggshell was easily identified with the naked eye, as well as under the SEM (Figs. 1C and 1D), the first round confirmed the detectability by Raman spectroscopy of the HAp of the cuticle layer (Figs. 1E–1F). Due to the incomplete preservation of the cuticle in the chicken eggshell, we performed Raman surface mapping on the area where we found possible cuticle remains using the SEM and EPMA previously.

Raman analysis II (referred to as RII hereafter) was performed using the Horiba Jobin Yvon GmbH LabRam Raman spectrometer equipped with a 632 nm laser. All results were obtained using the 10× objective of the integrated Olympus microscope. While the 10× objective is not strongly sensitive to rare compounds, it provides a larger field of view for faster qualitative detection of HAp on eggshell sample than in RI. In order to distinguish HAp and calcite from sediments, we collected Raman spectra of the fresh chicken eggshell, the studied fossil eggshells and their associated sediments (Fig. 2A). All Raman spectra of eggshells and their associated sediments during RII were imported into the statistics software R and normalized with the R package "ChemoSpec" (*Hanson, 2017*). We performed principle component analysis (PCA) on the spectral area of 500 to 1,500 cm$^{-1}$ using the same package (Fig. 2B).

## RESULTS

The cuticle layer on the *Gallus* eggshell was easily observed with the naked eye. The RI Raman spectrum from the *Gallus* eggshell shows a significant peak at 1,087 cm$^{-1}$ that arises from the calcite, which is the major component of eggshell (Fig. 1F). Another intense peak at 967 cm$^{-1}$ was assigned to the $v_3$ symmetric stretching vibrations (*Walters et al., 1990*; *Gergely et al., 2010*; *Frost et al., 2014*; *Igic et al., 2015*). Targeting the 967 cm$^{-1}$ peak, the Raman surface scan shows the patchy distribution of the HAp layer overlying the calcitic eggshell (Fig. 1E). The result demonstrates detectability of the inner HAp layer of the cuticle by RS.

On the outside of *Macroolithus yaotunensis* and *Triprismatoolithus stephensi* eggshell fragments, we also found peaks in these spectral regions (Fig. 2A). RII showed two intense broad peaks that are located between 1,063–1,097 cm$^{-1}$ and 972–986 cm$^{-1}$ in the spectra. The spectral range of 1,063–1,097 cm$^{-1}$ can be assigned to the C–O bond of calcite, which is observable in the spectra from all eggshells. The 972–986 cm$^{-1}$ spectral ranges arose possibly from the phosphate ($PO_4^{3-}$) or hydrogen phosphate ($HPO_4^{2-}$) (*Sauer et al., 1994*; *Crane et al., 2006*; *Igic et al., 2015*). Importantly, the 972–986 cm$^{-1}$ spectral ranges are present in the spectra from the chicken and dinosaur eggshells but absent in the ones from the crocodilian eggshells and surrounding sediments. The chemometric analysis using ChemoSpec suggests that both chicken and dinosaur eggshells show similar spectral pattern between 800 cm$^{-1}$ and 1,200 cm$^{-1}$ (Fig. 2B). In addition, the spectra from the crocodilian eggshells and surrounding sediments occupy different chemospaces from the ones of dinosaur and chicken eggshells (Fig. 2B).

Under the polarizing microscope, the *Macroolithus* eggshell from the Hongcheng Basin of Jiangxi, China, displayed distinct two crystalline layers overlain by an enigmatic layer

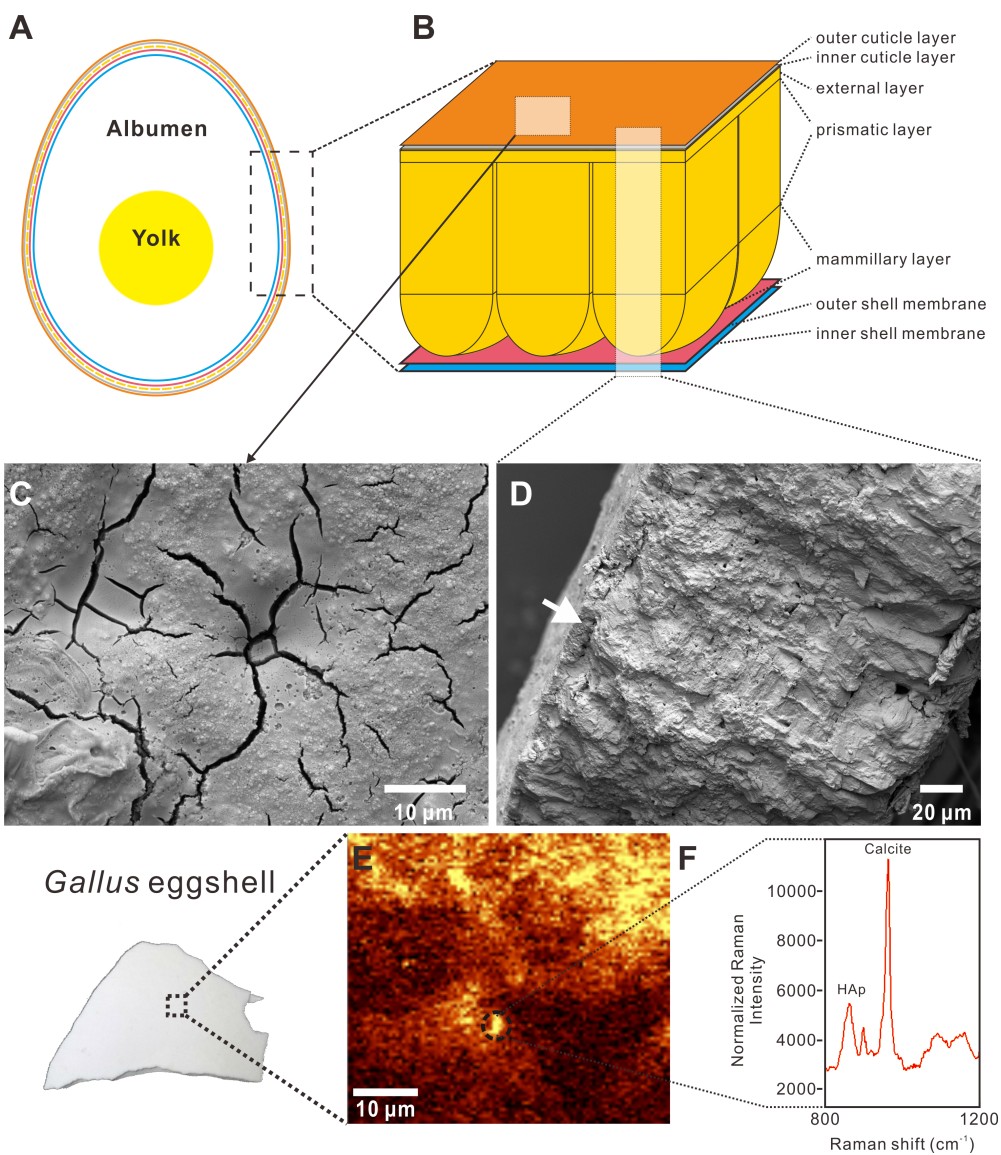

**Figure 1   Cross-sectional view, SEM images, and Raman imaging and spectrum of a *Gallus gallus domesticus* egg and eggshell.** The Raman image and spectrum were collected using 532 nm excitation wavelength and a 50x objective with RI. (A) The generalized anatomy of an egg. (B) The chicken eggshell comprises three crystalline layers, including the mammillary layer, prismatic layer, and external layer. The cuticle layer overlying the calcareous eggshell is further divided to two layers, including a HAp inner layer and a proteinaceous outer layer. The shell membrane, namely *membrane testacea*, is also characterized by two layers. (C) SEM image of the cuticle on the surface of the *Gallus* eggshell, showing a patchy and cracked pattern. (D) SEM image of the radial section of the *Gallus* eggshell. The white arrow indicates the cuticle layer that lies on the calcitic eggshell. (E) Raman chemical image with peak targeting at 967 cm$^{-1}$ which is attributed to HAp. The yellow area represents the patchy distribution of the inner HAp cuticle layer on the *Gallus* eggshell. The calcitic eggshell that is not covered by the inner HAp cuticle layer is shown in the brown area. The dotted circle corresponds to the spectrum shown in (F). (F) Spectrum collected in the dotted-circle area of the Raman chemical image shown in (E). Two significant peaks at 967 cm$^{-1}$ and 1,087 cm$^{-1}$ represent HAp and calcite, respectively. Credit of the SEM images and drawings: Tzu-Ruei Yang (Universität Bonn).

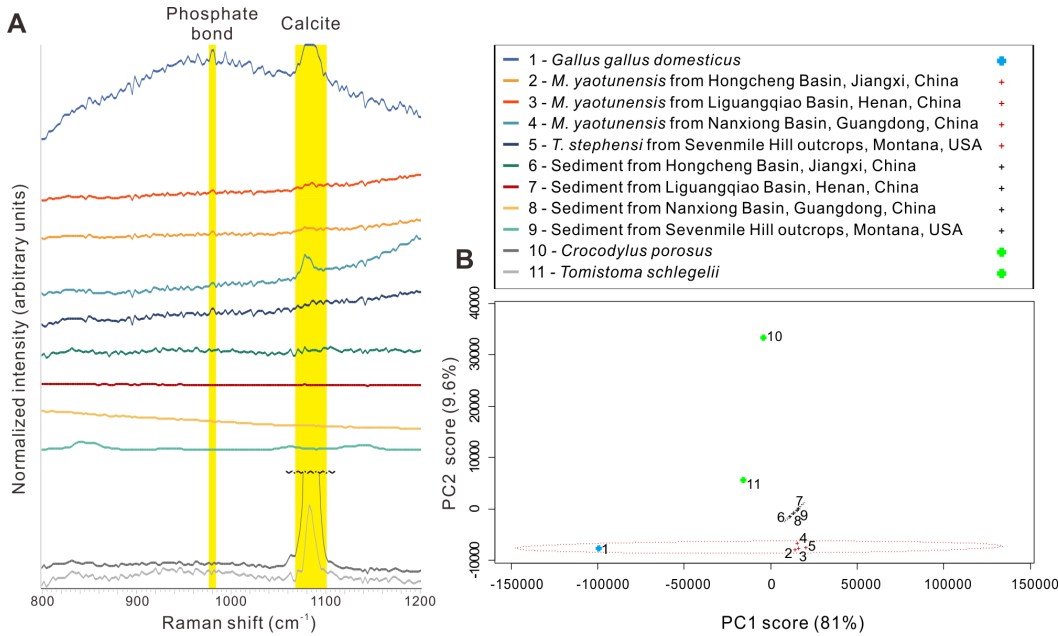

**Figure 2  Raman spectra and chemometric analysis.** (A) Raman spectra derived from chicken, crocodilian, fossil dinosaur eggshells, and surrounding sediments. The peaks around 972–986 cm$^{-1}$ and 1,063–1,097 cm$^{-1}$ are marked by yellow bars, indicating the calcite and phosphate bonds. (B) The principle component chemometric analysis on the spectral area of 800–1,200 cm$^{-1}$ shown in (A) demonstrates significant disparities between dinosaur/chicken eggshells, crocodilian eggshells, and sediments. Credit of the drawings: Tzu-Ruei Yang (Universität Bonn).

and surrounding sediment (Fig. 3A). These features are also present in the other two *Macroolithus* eggshells (see Supplemental Information). The uneven thickness of the prismatic layer in the Jiangxi *Macroolithus* eggshells was a result of unpolished natural fracture surface of the section (Fig. 3B). A possible patchy and flake-like cuticle layer was observed with SEM on the outer surface of the *Macroolithus* and *Triprismatoolithus* eggshells (Figs. 3C & 4B). Furthermore, an enigmatic protruding fiber-like structure was recognized in the *Triprismatoolithus* eggshell (Figs. 4C–4D). The diameter of the fiber-like structure is approximately 3 μm.

We performed ten elemental line-mappings on each eggshell with EPMA and targeted phosphorus of the HAp. Of the of total 30 line mappings on the three *Macroolithus yaotunensis* eggshell, only two of them, which were from the Jiangxi and Guangdong specimens, show a significant phosphorus peak on the boundary between the eggshell and surrounding sediment (Figs. 3D & 3F). The elemental analysis also indicates the scarcity of the phosphorus in the surrounding sediment. In addition, the P concentration increases from the mammillary layer to the prismatic layer in all analyzed oviraptorid eggshells (Figs. 3D–3F and raw data in the Supplemental Information).

In the EPMA line-scan on the *Triprismatoolithus stephensi* eggshell, three significant peaks for P were observed (Fig. 4E). The peak on the boundary between the eggshell and surrounding sediment indicates possible cuticle preservation (Fig. 4E). However, the

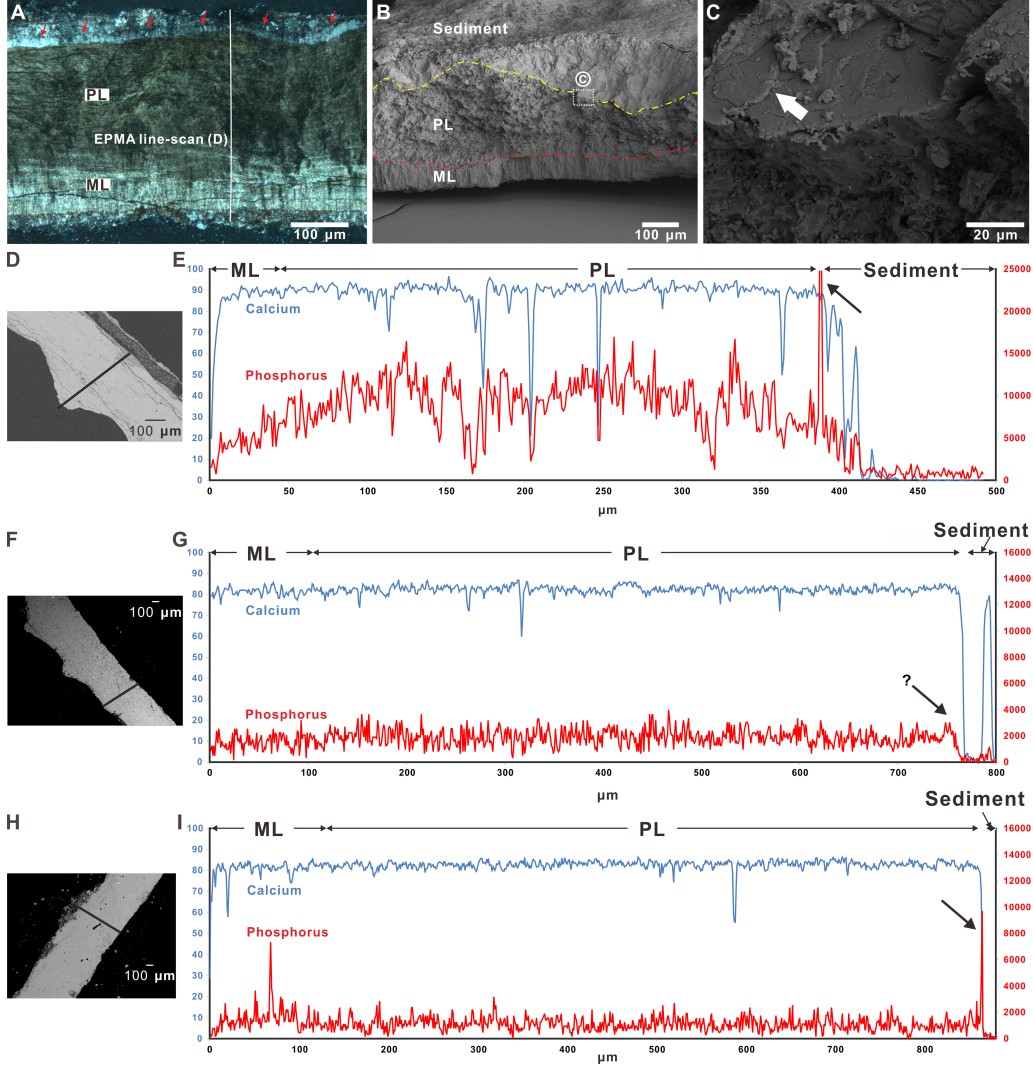

**Figure 3** Microscopic images of *Macroolithus yaotunensis* eggshell from Henan, Jiangxi, and Guangdong and their line-scan spectra. (A) The *Macroolithus yaotunensis* eggshell (NMNS CYN-2004-DINO-05-I) from Jiangxi under the polarized light microscope. The red arrows indicate the indistinct boundary between the sediment and an enigmatic layer. This enigmatic layer probably was formed by the interaction between organics of the egg and surrounding sediment. The location of the EPMA line-scan spectra in (D) is indicated by the white solid line. In this eggshell image, a smooth boundary between the prismatic and mammillary layers (PL and ML, seen as dark and light zones) is clearly observed. (B) SEM image of radial untreated fracture of the *Macroolithus yaotunensis* eggshell. The boundary between the sediment and prismatic layer is marked by yellow dashed line based on their distinct structural features. The boundary between ML and PL is marked by the red dashed line. The ML shows the distinct vertical mammillae structure. (C) Close-up image of the area in white box in (B). Possible preservation of cuticle is indicated by the white arrow, showing patchy and flaky structures similar to the cuticle on the modern chicken eggshell (Figs. 1C and 1D). (D, F, H) The microscopic image shows the radial cut section of studied *Macroolithus yaotunensis* eggshells seen in the SEM coupled with the EPMA and elemental line-scan track marked as black lines. 

**Figure 3 (…continued)**
(D) The spectra illustrate the distribution of Ca and P across the eggshell from the Hongcheng Basin in Jiangxi shown in (A), from innermost (left) to outermost, as indicated by the white line in the microscopic image. The Ca spectrum clearly marks the extent of the calcitic eggshell. A significant peak that is marked by a black arrow demonstrates a relatively high concentration of P, indicating the possible preservation of the cuticular HAp layer. (G) The spectra illustrate the distribution of Ca and P across the eggshell from the Liguangqiao Basin in Henan, from innermost (left) to outermost. The Ca spectrum clearly marks the extent of the calcitic eggshell. A slightly higher amount of P near the interface between the eggshell and sediments, as indicated by a black arrow, is observed. (I) The spectra illustrate the distribution of Ca and P across the eggshell from the Nanxiong Basin in Guangdong, from innermost (left) to outermost. A significant peak that is marked by a black arrow demonstrates a relatively high concentration of P, indicating the possible preservation of the cuticular HAp layer. ML, mammillary layer; PL, prismatic layer. Credit of the SEM images and drawings: Tzu-Ruei Yang (Universität Bonn).

other two peaks were seen in the sediment and on the boundary between the external layer and prismatic layer. The variation of P concentration between different eggshell layers was also observed in *Triprismatoolithus stephensi* eggshell (Fig. 4E and Supplemental Information). Although the P concentration within the mammillary and prismatic layers shows little variation, the external layer has a significantly lower concentration of P than the mammillary and prismatic layers (Fig. 4E & raw data in the Supplemental Information).

# DISCUSSION

## Elemental analysis of theropod eggshells

While previous studies have reported possible preservation of cuticle and *membrana testacea* (*Chow, 1954*; *Kohring & Hirsch, 1996*; *Grellet-Tinner, 2005*; *Jackson & Varricchio, 2010*; *Reisz et al., 2013*; *Yang et al., 2015*), none of these studies employed chemical analyses to test this hypothesis. In this study, elemental analysis suggests the preservation of the inner hydroxyapatitic cuticle layer based on the high concentration of P at the boundary between eggshell and sediment. However, the peak that corresponds to a high concentration of P is not always present in each elemental line-scan. For instance, on the *M. yaotunensis* eggshell sample from the Liguangqiao Basin (Fig. 3E), a probable P-rich zone, instead of an abrupt peak, is shown near the boundary between the eggshell and surrounding sediment. RS analysis indeed revealed the existence of phosphates. Since RS scans a larger area than EPMA does, we suggest that the absence of an abrupt peak for a P-rich zone in those 28 line-scans is a result of incomplete preservation of the cuticle layer or a sampling bias during the EPMA analyses. The elemental analysis also shows that P is rare in the surrounding sediment from both the Nanxiong Group of China and the Two Medicine Formation of Montana, United States. The deficiency of P in the sediment thus argues against an allochthonous origin of the P in the outermost part of the eggshell (Figs. 3D–3F & 4E). The chemometric comparison of the spectra derived from sediments also suggests the depletion of phosphate minerals in the sediments. Therefore, the peak in the sediment surrounding the *Triprismatoolithus stephensi* eggshell might be derived from the eggshell specimen itself.

The enigmatic fiber-like structure near the base of the mammillary layer in the *Triprismatoolithus* eggshell (Fig. 4D) may be a capillary vessel of the chorioallantoic

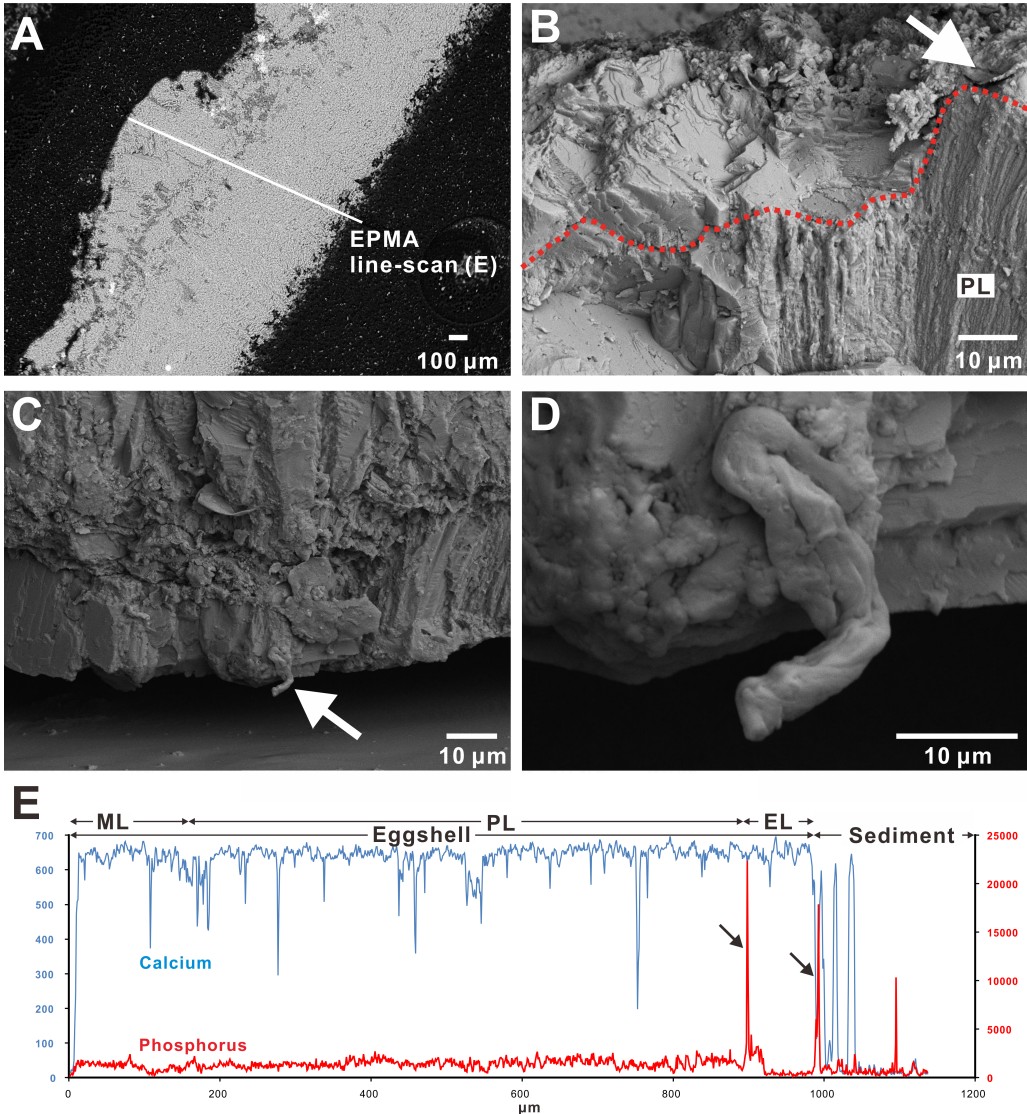

**Figure 4  SEM images of the *Triprismatoolithus* eggshell and EPMA line-scan spectra.** (A) The radial cut section of studied *Triprismatoolithus* eggshell in the SEM coupled with the EPMA. (B) The probable flake-like cuticle structure atop the eggshell, marked by the white arrow. The red dotted line marks the outermost boundary of the prismatic layer. (C) An enigmatic structure (white arrow) protruding from the mammillary layer, possibly a fiber of the *membrana testacea* extending into the mammillary layer. (D) Enlargement of the possible fiber. (E) Line scans across eggshell in (A). The scan shows a high concentration of P at the boundary between the eggshell and surrounding sediment. Two other peaks of P were observed on the boundary between the external and prismatic layer and in the sediment. The P concentration does not differ between the mammillary and prismatic layers; however, the external layer showed a significantly lower concentration of P than the mammillary and prismatic layers. EL, external layer; ML, mammillary layer; PL, prismatic layer. Credit of the SEM images and drawings: Tzu-Ruei Yang (Universität Bonn).

membrane or a fiber of the *membrana testacea*. The elemental analyses cannot provide evidence for the preservation of the capillary vessels of the chorioallantoic membrane. SEM images show that the vessels in the chorioallantoic membrane of modern chicken eggshells range in size from 3 to 10 μm (*Maiber et al., 2016*), similar to the fiber-like structure we observed in the *Triprismatoolithus* eggshell (Fig. 4D). However, the fiber-like structure in the *Triprismatoolithus* eggshell does not appear to be hollow, unlike a blood vessel (*Schweitzer et al., 2005*). It is therefore possible that the enigmatic fiber-like structure in the *Triprismatoolithus* eggshell represents a part of fossilized *membrana testacea*, which has never been described in theropod dinosaurs. The *membrana testacea* fibers in a chicken egg occasionally penetrate through the mammillary base (*Ahmed, Suso & Hincke, 2017*) and are of the same size of around 3 μm as the fiber-like structure we observed in the *Triprismatoolithus* egg (Fig. 4D). However, the hypothesis of fossilized *membrana testacea* requires more supporting evidence.

## Variation of P concentration in eggshells

Despite numerous studies on element distribution in modern avian and reptilian eggshells, dinosaur eggshells were rarely investigated from a chemical perspective. Several studies posited that P is a critical element that regulates eggshell growth (*Fink et al., 1993*; *Dennis et al., 1996*; *Fraser, Bain & Solomon, 1999*; *Cusack, Fraser & Stachel, 2003*). In *Gallus gallus domesticus* eggshells, a pattern of increasing P concentration from the mammillary layer to the prismatic layer has been reported (*Cusack, Fraser & Stachel, 2003*). This pattern is similar to what we observed in oviraptorid eggshells (Figs. 3D–3F). The deficiency of P in the surrounding sediments demonstrates that the distribution pattern of P in the studied eggshells is endogenous. Since ornithological research had suggested that P is an inhibitor of calcite growth during eggshell formation (*Fink et al., 1993*; *Dennis et al., 1996*; *Fraser, Bain & Solomon, 1999*; *Cusack, Fraser & Stachel, 2003*), the similarity of P distribution between oviraptorid dinosaur and modern avian eggshells suggests that the oviraptorid dinosaurs studied by us already possessed a similar eggshell formation mechanism to that of extant birds.

Moreover, *Cusack, Fraser & Stachel (2003)* also observed that the P concentration increases more sharply between the mammillary and prismatic layers in the eggshell of younger birds than in those of older birds. Therefore, the difference in P distribution between young and older birds can probably help paleontologists decipher whether an oviraptorid clutch was laid by a single female or by several females. If the whole clutch was produced by a single female or by several females of a similar age, all eggs should exhibit a similar pattern of P concentration increase from the mammillary layer to the prismatic layer. Conversely, different patterns of increase would suggest that females of different ages contributed to a single clutch. *Varricchio et al. (2008)* compared the ratio of clutch mass to adult mass and proposed that an oviraptorid clutch was produced by several females. *Yang et al. (2016)* have provided preliminary supportive results based on statistical analyses of eggshell P distribution and external morphology in the same oviraptorid clutch.

On the other hand, in the *Triprismatoolithus stephensi* eggshell, P concentration varies slightly between the mammillary and prismatic layer but decreases significantly from the

prismatic layer to the external layer (Fig. 4E). Such a pattern of P distribution differs from the one in oviraptorid or modern bird eggshells we discussed previously. Moreover, unlike a single peak of high P concentration in oviraptorid and modern eggshells, three distinctive peaks of high P concentration are present in the *Triprismatoolithus stephensi* eggshell. Based on the general deficiency of P in the surrounding sediment, we posit that the P peak in the sediment surrounding the *Triprismatoolithus stephensi* eggshell is a result of a detached cuticle layer. The peak at the boundary between the external layer and surrounding sediment represent the putative cuticle layer (Fig. 4E). However, the peak at the boundary between the prismatic and external layers was not reported previously in any avian eggshells (Fig. 4E). If the sequestration of P is a crucial factor in the termination of calcite growth, the high concentration of P on the boundary between the prismatic and external layers represents a possible temporal hiatus between the prismatic and external layers in the *Triprismatoolithus* eggshell.

## Interpretation of Raman spectra

In the Raman study of the fossil eggshells, the peak at 967 cm$^{-1}$ of *Gallus* eggshell was not seen in all spectra, but a broad peak ranging from 972–986 cm$^{-1}$ was detected instead (Fig. 2A). The broad peak ranging from 972–986 cm$^{-1}$ represent the phosphate groups, as indicated in *Igic et al. (2015)* (see also Supplemental Information). Three hypotheses can be formulated to explain the different spectral patterns from 960 cm$^{-1}$ to 980 cm$^{-1}$ in extant cuticle vs. the fossil eggshells.

First, theropod dinosaurs might have used whitlockite, a phosphate mineral containing hydrogen phosphate anions ($HPO_4^{2-}$), to form the inner cuticle layer instead of HAp. Whitlockite is the second most abundant mineral in biological calcified tissues, accounting for around 20% of bone mineral (*Driessens & Verbeeck, 1990*; *Jang et al., 2014*; *Jang et al., 2015*). It is also possible that the HAp of the inner cuticle layer is in a non-stoichiometric form, and hence part of the phosphate anion ($PO_4^{3-}$) is occupied by hydrogen phosphate ($HPO_4^{2-}$) (*Rey et al., 2011*; *De La Pierre et al., 2014*).

Second, the original HAp of the inner cuticle layer could have been transformed to whitlockite. *Jang et al. (2014)* showed that HAp transforms into dicalcium phosphate dihydrate (DCPD, $CaHPO_4 \cdot 2H_2O$) at 70 °C. The DCPD is then converted into whitlockite ($Ca_9(MgFe)(PO_4)_6PO_3OH$) by incorporating $Mg^{2+}$ or $Fe^{2+}$ when the pH decreases. In natural environments, such as the Nanxiong Formation and Two Medicine Formation, burial and diagenetic temperatures over 70 °C are unlikely except by an intrusion of igneous origin. However, geological investigations did not provide any evidence for such igneous intrusions in the Nanxiong Basin (*Erben, 1995*). At the Sevenmile Hill outcrops in Montana, United States, *Foreman et al. (2008)* reported an altered volcanic ash layer (bentonite) that is depleted in P. However, the volcanic ash layer was deposited prior to the eggshells (*Jackson & Varricchio, 2010*).

Third, the P we detected might be derived from the surrounding sediments during diagenesis. This is inconsistent with the preservation of pigments in the Nanxiong Basin eggshells and absence of igneous intrusions in both localities, which suggests that the

eggshells we studied experienced very little diagenesis (*Erben, 1995*; *Foreman et al., 2008*; *Wiemann et al., 2017*).

## Pigmentation on theropod eggshells

As noted above, the discovery of protoporphyrin (*Wiemann et al., 2017*) in oviraptorid dinosaur eggshells from all three localities where the eggshells for this study were collected (Table 1) is consistent with the cuticle preservation. Although *Wilson et al. (2017)* indicated that pigment deposition and cuticle formation are not codependent, the preserved protoporphyrin pigments was probably from the cuticle layer.

*Thomas et al. (2015)* performed Raman spectroscopy and mass spectrometry to show the detectability of pigments in extant avian eggshells. Our Raman analyses also demonstrate detectability of pigments in fossil eggshells; however, the peaks in the spectra do not completely correspondent with the ones shown in *Thomas et al. (2015)*, suggesting further Raman studies using mass spectrometry on dinosaur eggshells.

## Nesting ecology of oviraptorid dinosaurs

As noted, two nesting modes are recognized in birds, open and buried nesting (*Varricchio & Jackson, 2016*). Most birds build open nests either on the tree or on the ground (ostriches or emus), while the megapode birds bury their eggs in a mound. Therefore, all open-nesting birds have eggshells of low porosity (*Tanaka, Zelenitsky & Therrien, 2015*). However, an intermediate mode, semi-open nests in oviraptorid dinosaurs, was recently proposed based on detection of pigments and porosity evaluation (*Yang et al., 2015*; *Wiemann et al., 2017*). Pigmented eggshells strongly suggest an open nesting mode. However, the porosity evaluation revealed a heterogeneous distribution of porosity—highest porosity in the lower middle part (buried) and lowest porosity in the blunt end (open) (*Yang et al., 2015*; *Wiemann et al., 2017*), thus indicating a semi-open nesting mode. This intermediate mode is consistent with the fluvial environment inferred by sedimentological investigations in the Nanxiong Basin (*Erben, 1995*). In such a humid fluvial environment, the risk of microbial invasions was much higher. The cuticular nanospheres, which are associated with humid environments and protect extant eggs from microbial invasions, were likely present in the oviraptorid and alvarezsaurid dinosaur eggs from these fossil humid environments as well. Although we did not detect cuticular nanospheres in the dinosaur eggshell, our detection of a cuticle layer suggests it to have been an adaptation the oviraptorid and alvarezsaurid dinosaurs nesting in a humid fluvial environment (*D'Alba et al., 2016*).

On the other hand, it was inferred that Mongolian oviraptorid clutches were deposited in a xeric environment. The evidence is the eolian nature (sand dune deposits) in the Baruungoyot and the Djadokhta formations (*Norell et al., 1995*). Although no Mongolian oviraptorid eggshells were included in our study, it appears likely that Mongolian oviraptorid eggs were also cuticle-coated for the prevention of water loss as in Adélie penguin eggs (*Thompson & Goldie, 1990*). *Fanti, Currie & Demchig (2012)* documented clutches of the oviraptorid *Nemegtomaia* from both the Nemegt and Baruungoyot formations, that represent mesic and xeric habitats, respectively. Moreover, they also provided evidence for the lateral coexistence of mesic and xeric habitats in the Baruungoyot

Formation and posited that *Nemegtomaia* and other Mongolian oviraptorid dinosaurs nested near permanent or seasonal streams. In such a variable environment, the cuticle-coated eggs, therefore, would have been an adaptive trait that enhanced reproductive success of oviraptorid dinosaurs.

While it is uncertain that the alvarezsaurid eggshells were pigmented without further chemical studies, it is possible that alvarezsaurid dinosaur exploited similar nesting strategies as oviraptorid dinosaurs did.

### Taphonomic implications for cuticle preservation

The cuticle layer is a labile, easily lost structure that appears unlikely to be preserved during the fossilization process. Our study thus raises an important taphonomic question, i.e., how did the cuticle layer get preserved? Since the cuticle preservation observed by us appears to pertain to eggshell from clutches and not from dispersed eggshell, cuticle preservation probably is facilitated by fossilization of eggs in clutches. Furthermore, a layer of secondary diagenetic calcite is commonly observed on eggshells from fluvial deposits (Fig. 3A; *Jackson & Varricchio, 2010*). This layer of secondary calcite may have been a rigid shield that protected the cuticle layer from mechanical removal.

Therefore, we hypothesize that the *in situ* preservation and the coverage of secondary calcite possibly protect the cuticle layer from being washed away or removed. More detailed microscopic observation of the association between the cuticle, secondary calcite layers, and surrounding sediment will possibly elucidate the taphonomic processes.

## CONCLUSIONS

Our study suggests preservation of the inner cuticle layer in *Macroolithus yaotunensis* and *Triprismatoolithus stephensi* eggs based on elemental analysis and RS. The cuticle layer and a semi-open nesting mode in humid environments may be key adaptations to protect the eggs from microbial invasions in a humid environment such as a fluvial plain, as does the cuticle layer of extant birds nesting in humid environments. In combination with the previous discovery of pigments in oviraptorid dinosaur eggshells, this study provides further evidence for preservation of soft tissues in the fluvial deposits, especially in the Cretaceous red bed basins of China. Further chemical studies on dinosaur eggshells will shed additional light on the reproductive biology of dinosaurs, as well as on taphonomic questions.

## INSTITUTIONAL AND TECHNICAL ABBREVIATIONS

| | |
|---|---|
| EPMA | Electron probe microanalysis |
| MOR | Museum of the Rockies, Bozeman, Montana, U.S.A |
| MPIE | Max-Planck-Institute for Iron Research, Dusseldorf, Germany |
| NMNS | National Museum of Natural Sciences, Taichung, Taiwan |
| RS | Raman spectroscopy |
| SEM | Scanning electron microscopy |
| STIPB | Bereich Paläontologie, Steinmann-Institut für Geologie, Mineralogie und Paläontologie, Universität Bonn, Bonn, Germany |

| TEM | Transmission electron microscopy |
| WDS | Wavelength dispersive spectrometer |
| ZAF | Matrix correction factors: Z, atomic number correction; A, absorption correction; F, characteristic fluorescence correction |

## ACKNOWLEDGEMENTS

The SEM observations and EPMA analyses were performed at the Steinmann-Institute of the University of Bonn, Bonn, Germany. Prof. Dr. Andreas Erbe and Ms. Petra Ebbinghaus kindly provided the access to the WItec Raman and Horiba Jobin Yvon Raman spectrometers at the MPIE. Fossil eggshell samples were provided by Prof. Dr. David Varricchio at the Montana State University, Bozeman, MT, USA. Dr. Georg Heumann at the STIPB, and Prof. Dr. Yen-Nien Cheng at the NMNS. This is contribution number 1 of DFG Research Unit 2685 ''The Limits of the Fossil Record: Analytical and Experimental Approaches to Fossilization.'' The authors would also like to thank Daniel Lawver and two anonymous reviewers for their constructive comments.

### Funding

This research is supported by the scholarship for studying abroad from the Ministry of Education, Taiwan and research grant from the Jurassic Foundation to Tzu-Ruei Yang. Tzu-Ruei Yang and P. Martin Sander are both funded by the Deutsche Forschungsgemeinschaft (DFG, German Research Foundation)—project number 348043586. The funders had no role in study design, data collection and analysis, decision to publish, or preparation of the manuscript.

### Grant Disclosures

The following grant information was disclosed by the authors:
Ministry of Education, Taiwan.
Jurassic Foundation.
Deutsche Forschungsgemeinschaft: 348043586.

### Competing Interests

The authors declare there are no competing interests.

### Author Contributions

- Tzu-Ruei Yang conceived and designed the experiments, performed the experiments, analyzed the data, contributed reagents/materials/analysis tools, prepared figures and/or tables, authored or reviewed drafts of the paper, approved the final draft.
- Ying-Hsuan Chen and Beate Spiering performed the experiments, contributed reagents/materials/analysis tools, approved the final draft.
- Jasmina Wiemann conceived and designed the experiments, analyzed the data, authored or reviewed drafts of the paper, approved the final draft.

- P. Martin Sander conceived and designed the experiments, contributed reagents/materials/analysis tools, authored or reviewed drafts of the paper, approved the final draft, manuscript editing.

## Data Availability

The raw data are provided in the Supplemental Files.

## Supplemental Information

Supplemental information for this article can be found online at http://dx.doi.org/10.7717/peerj.5144#supplemental-information.

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
