# Peer review of "Fossil eggshell cuticle elucidates dinosaur nesting ecology"

_PeerJ, doi:10.7717/peerj.5144_

## Round 0.1 · original submission · Minor Revisions

Dear authors,

We have now three review reports about your manuscript. They recommend publication after minor revision, but their concerns would include the improvement and revision of some important issues that may reflect weakness in your conclusions. Thus, please, consider all of them carefully.

I have seen also some troubles considering your interpretations of the functions of the cuticle and pigments and the meaning of its probable presence in some dinosaurs. Note that they are in consonance with concerns from Reviewer 3.

For instance,

1-The control of the humidity is dependent mostly of the nest type (see Deeming 2011), and this is valid to birds as well as to other living archosaurs (e,g. crocodiles) (see Tanaka and Selenitsky, 2013).

2-There is no codependence of the deposition of pigments and cuticle (Wilson et al., 2017). The pigments are not exclusive to the cuticle layer, but they are mostly present in the calcareous layers. Wilson et al., 2017 seem to have demonstrated that the pigments are deposited previously to the cuticle. Thus, it appears to be a lot of controversy around this issue and must be reflected in the manuscript

3- Apparently, there indeed are also some similitude in the formation of avian and crocodile eggshell. See the following paragraph extracted from Mikšík et al., 2017:

“Substantial differences between proteins in the eggshell of crocodile and previously described birds’ eggshells exist (both in terms of quality and quantity), however, the entire proteome of Crocodilians has not been described yet. The most abundant protein was thyroglobulin. The role of determined proteins in the eggshell of the Siamese crocodile is discussed. For the first time, the presence of porphyrin pigment is reported in a crocodilian eggshell, albeit in a small amount (about 2 to 3 orders of magnitude lower than white avian eggs)” (Mikšík et al., 2017)
4-Other aspect that I think should be considered is the fact that the cuticle is easily broken off, even washing of the egg makes it disappear. So, I was wondering if you can explain which processes could be acted to preserve so much delicate structure in an inferred fluvial environment.

I hope you find useful the reviewers comments and suggestions, as well as those from me, to improve your manuscript and resubmit it soon.

Kind regards,
Graciela Piñeiro

Reviewer 1 ·

Basic reporting

Although authors intend to highlight a possible boundary between the sediment and an enigmatic layer in fig.3A, significant peaks of P in fig. 3D, E, F and fig.4E, the white dashed line and yellow bars totally cover the elements they want to show. I suggest they use arrows instead.

Experimental design

No comment

Validity of the findings

Authors argued that "the P concentration increases from the mammillary layer to the prismatic layer but decreases from the prismatic layer to the external layer" in line 262-263, but said "the P concentration is homogeneous throughout the mammillary and prismatic layers, but decreases in the external layer"in the note of fig.4E.

Additional comments

This article is really interesting for authors have found a chemical method to detect the existence of cuticle layer in fossil eggs. I am looking forward to seeing more chemical methods which can solve the issues in dinosaur nesting ecology.

·

Basic reporting

No comment.

Experimental design

No comment.

Validity of the findings

No comment.

Additional comments

The manuscript entitled “Fossil eggshell cuticle elucidates dinosaur nesting ecology” by Yang et al. presents chemical analyses suggesting the preservation of eggshell cuticle in non-avian dinosaur eggs from China and North America. I find that this research is deserving of publication and requires only minor revisions before final acceptance by the journal.

While I find the experimental design to be appropriate, I would suggest the inclusion of a negative control. Chemical analysis of a modern reptilian eggshell will provide an appropriate comparison for fossil specimens and will support the conclusions found in the study. Specific questions that I have include, is a phosphorus peak present on modern eggshells that do not contain a cuticle layer and if so, what does this imply for fossil specimens?

I also believe that the main conclusions of this research are appropriate. However, I do not agree with the conclusion that the “tube-like” structure found at the inner eggshell surface of one of the fossil specimens is a capillary vessel associated with the chorioallantoic membrane. Similar structures have been mentioned previously in the literature and tentatively identified as fibers of the outer eggshell layer but never as a capillary vessel. Additionally, Figure 4 does not demonstrate that this structure is hollow and therefore, not “tube-like”. This is in contrast to preserved capillary structures found in non-avian dinosaurs by Schweitzer’s lab, in which vessels are preserved in three-dimensions and are hollow. I would suggest listing all possible structures that this item could be identified as or provide more evidence as to why it is or isn’t a capillary vessel.

Finally, while the manuscript is generally well written, I have made some minor suggestions that should improve the grammar and legibility of the paper. See attached pdf for suggested edits.

Reviewer 3 ·

Basic reporting

See general comments

Experimental design

no comments

Validity of the findings

See general comments

Additional comments

Comments on manuscript #19998
Yang et al.: Fossil eggshell cuticle elucidates dinosaur nesting ecology


In this study, the authors used for the first time a combination of microscopic and chemical techniques (SEM, EPMA, Raman Spectroscopy) to test for the presence of a cuticle layer in the eggshells of 4 non-avian dinosaurs eggs.
This is a very interesting study and the manuscript is well written. They present several evidences for the preservation of a cuticle layer in these non-avian dinosaur eggs.
However, as mentioned in the comments below, some of their statements are vague or inexact. Their discussion regarding the dinosaur nesting ecology should be further developed. The authors should nuance some of their interpretations on the nesting ecology of non-avian dinosaurs.

In summary, there are some points in the introduction and the discussion that should be clarified/ developed before acceptance of the manuscript.


Comments

Introduction, L 56: “In addition to lipids, the avian cuticle contains proteins, polysaccharides, calcium carbonate (vaterite), calcium phosphates (hydroxyapatite, HAp)…(…)…. Among these components, the vaterite and HAp in avian cuticle are stored in the form of nanospheres”.
To my understanding, avian cuticle always contains lipids, proteins, and polysaccharids. However, the nano-spheres of vaterite and hydroxyapatite are not always present (e.g., D’Alba et al., 2016). You should thus be more precise in your statement here.

Introduction, L 64: “Basically, the cuticle is divided into two distinct layers, including an amorphous HAp inner layer and a proteinaceous outer layer (Simons, 1971; Dennis et al., 1996; Fraser, Bain & Solomon, 1999)”. Is that always the case? Some bird cuticles do not comprise Hap nano-spheres (e.g., D’Alba et al., 2016).

Introduction, L 83: “Conversely, avian eggshells have HAp granules along the inner periphery on calcareous eggshells, which are absent in reptilian eggshells”. Again, you should be more precise with your statements. The structure and chemical compositions of cuticles greatly vary between bird species. Only some avian species present Hap granules in their cuticle. Some species present vaterite granules instead. Some species do not show these mineral nano-spheres at all (e.g., D’Alba et al., 2016).

Introduction, L86: “to map the P and calcium distribution”. Precise at least once that “P” refers to phosphorus.

Introduction, L86: “Kusada et al. (2011) applied electron microscopy to map the P and calcium distribution in several extant avian eggshells and discovered that P is confined to the cuticle layer. Hence, P is a distinct marker for identifying the cuticle layer in case of P depletion in sediment.” You should precise that the phosphorus concentration in the cuticle varies greatly between bird species and that phosphorus can potentially be absent (cf. Pelican specimen in the study of Kusada et al., 2011).

Introduction, L89: “Although it is still not clear how P functions during the eggshell formation, a study suggested that P terminates eggshell growth since it competes with calcium during the eggshell crystallization”. You could maybe be more precise and say something like “the inclusion of P stops the eggshell growth”…

Introduction, L 117: “Wiemann et al. 2017”. For consistency, there is a comma missing after “et al.” Please check and modify, if necessary, all other citations.

Introduction, L 122: “Among all dinosaurs, the nesting type of titanosaur, troodontid, and oviraptorid dinosaurs were comparatively well documented”. Provide some citations here.

Introduction, “Dinosaur nesting ecology”: In this section I would expect to learn more about the difference in cuticle structure between the megapodes and other birds and see if there is a direct relationship between the cuticle structure or the presence/absence of cuticle and the type of nest (open vs. buried) and the environment (dry vs. humid). This is not detailed enough. You should further present the studies of d’Alba et al. (2016, 2017) that have shown that the presence of vaterite or hydroxyapatite nano-spheres in the cuticle of some bird species is likely related to the environment (warm and/or humid) and the nest type (on the ground or mount).

Introduction, L 134: “has been remained uncertain". This is not correct English. You could rather say “has remained uncertain”.

Introduction, L144: “The goal of this study is to present further evidence of the cuticle preservation on dinosaur eggshells by means of the elemental analysis and Raman microscopy”. You should also mention the use of SEM imaging.

Introduction, L148: “Besides, nesting on the ground or in mounds increase the risk of microbial invasions, a cuticle layer should hence be prevalent on eggshells of semi-open nesting theropods”. Provide citations here.

Material and Methods, L 157: “We sampled three oviraptorid eggshells that were collected from three localities in China (Henan Province, Jiangxi Province, and Guangdong Province)”. You should precise here that the three localities are all part of the Nanxiong Group.

Material and Methods, L 158: “were housed at the NMNS and STIPB". To which museums do these abbreviations correspond? Give the full museums’ names or refer to the supplementary information file.

Discussion, L 267: “While previous studies have reported possible preservation of cuticle and membrane testacea”. The membrane testacea studies were not mentioned in the introduction. Provide a brief description of this structure.


Discussion, L270: “In this study, elemental analysis suggested the preservation of the inner hydroxyapatitic cuticle layer based on the high concentration of P at the boundary between eggshell and sediment.” You should also discuss the “negative” results that you obtained. You mentioned in the results section that “Of in total 30 line-mappings on the three Macroolithus yaotunensis eggshell, only two of them, which were from the Jiangxi and Guangdong specimens, show a significant peak for rich phosphorus on the boundary between the eggshell and surrounding sediment”. How do you explain that only 2 out of 30 line-mappings done on the Macroolithus yaotunensis specimens show a significant peak for phosphorus at the boundary between the eggshell and the sediment? You should discuss that.

Discussion, L281: “(Maiber et al., 2015)”. This reference is listed as Maiber et al., 2016 in the references list.

Discussion, L283: “However, no features of chorioallantoic membrane was observed. The preservation of the vessel in the chorioallantoic membrane is hence pending for more supporting evidence”. Specify what would be features of the chorioallantoic membrane. Also, how could you test for the presence of vessels of the chorioallantoic membrane?

Discussion, L288: “The elemental analyses show that P concentration increases from the mammillary layer to the prismatic layer of both Macroolithus yaotunensis and Triprismatoolithus stephensi eggshells (Figs. 3D-F & 4E)”. This statement is not observable on Fig. 3D-F, because the mammillary layer is not represented on these spectra. This is not clear either on Fig. 4E. You should maybe refer to the raw data.

Discussion, L296: “Moreover, their study also revealed that the P concentration increases more sharply in the eggshell of younger birds than in older birds.” Increase in P concentration between the mammillary layer and the prismatic layer?? Precise.

Discussion, L302: “Furthermore, the difference in P distribution between young and older birds can probably help paleontologists decipher if an oviraptorid clutch was laid by a single female or several females”. Why? Because females would be expected to have different ages? You should elaborate on that.

Discussion, L312: “Three hypotheses can be formulated to explain the different spectral patterns from 960 cm-1 to 980 cm-1 in extant cuticle vs. the fossil eggshells”. Where do you get these values from? Explain and cite Supplementary Information if needed.

Discussion, L345: “Thomas et al. (2016)” is listed as Thomas et al. (2015) in the references list.

Discussion, L351: You should develop much more the discussion on the nesting ecology of the sampled dinosaurs. I would suggest, for example, to explain in a few sentences, how eggshell porosity and eggshell color relate to the nest type.

Discussion, L355: “However, an intermediate mode — semi-open nests in oviraptorid dinosaurs was proposed based on detection of pigments and porosity evaluation (Yang et al., 2015; Wiemann et al., 2017).” Develop a bit on that. Why does egg pigmentation suggests an open nest in oviraptorid dinosaurs?

Discussion, L360: “Our detection of a cuticle in these eggs also supports this nesting ecology since the cuticle layer hinders microbial invasions (D'Alba et al., 2016)”. Develop on that. D’Alba et al., 2016, 2017 papers have shown that this is more specifically the mineral nano-spheres present in the cuticle of some bird species that help reduce microbial invasion in humid/warm environments. The cuticle itself is of course an important component of the antimicrobial defense, but its presence is not necessarily indicative of a humid/warm environment. A cuticle can be present in bird species nesting in very different environments.

Discussion, L362: “However, it was suggested that Mongolian oviraptorid clutches were discovered in the sand dune deposits (eolian) such as Baruungoyot Formation or Djadokhta Formation (Norell et al., 1995). Given the arid sand deposits, the cuticle layer was hence unlikely present on Mongolia oviraptorid eggshells, superficially contradicting to the inferences of this study.” Not necessarily. It has been shown that some bird species living in very arid environments present a relatively thick cuticle. This is for example the case of the Adelie penguin living in the very arid Antarctic environment. These birds present a very thick cuticle that helps reducing the rate of water loss of the egg (Thompson & Goldie, 1990).

Discussion, L363: Norell et al., 1995 is missing in the references list.

Conclusions, L376: “Our study suggests preservation of the inner cuticle layer in eggs of the oviraptorid Macroolithus yaotunensis”. You should rather say “in Macroolithus yaotunensis eggs” or “in eggs assigned to the oviraptorid Heyuannia huangi”.

Conclusion, L378: “The potential co-evolution of a cuticle layer and semi-open nesting in humid environments may have allowed theropod dinosaurs to nest in fluvial environments, as does the cuticle layer of extant birds nesting in humid environment”. Again, the presence of a cuticle is not necessarily indicative of a humid environment in modern birds (see D’Alba et al., 2016, 2017).

Figure caption 1F: “Two significant peaks at 967 cm-1 and 1087 cm-1 are indicated by yellow bars, representing HAp and calcite, respectively”. The yellow bars are actually not visible on the spectrum.

Figure caption 2A: “Raman spectra derived from chicken, fossil eggshells, and surrounding sediments. The peaks around 972-986 cm-1 and 1063-1097 cm-1 are marked by yellow bars, indicating the calcite and phosphate bonds”. You could actually indicate directly on the spectrum that the first peak represents the phosphate bounds and that the second peak represents the calcite bounds.

Figure caption 3: “Microscopic images of Heyuannia huangi eggshells”. To avoid confusion, and be consistent with the rest of the figures/text, you should rather say “Microscopic images of Macroolithus yaotunensis eggshells”.

Figure 3A: “EPMA Line-Scan (B)” is indicated on the image. Shouldn’t it be “EPMA Line-Scan (D)”?

Figure 4A: you should orientate the cross-section. Which side corresponds to the surface of the eggshell (with the cuticle layer)?

Table 1: “Oospecies” is misspelled.

Literature cited:
D'Alba, L., Maia, R., Hauber, M. E., & Shawkey, M. D. 2016. The evolution of eggshell cuticle in relation to nesting ecology. Proc. R. Soc. B, 283(1836), 20160687.
D’Alba, L., Torres, R., Waterhouse, G. I., Eliason, C., Hauber, M. E., & Shawkey, M. D. 2017. What Does the Eggshell Cuticle Do? A Functional Comparison of Avian Eggshell Cuticles. Physiological and Biochemical Zoology, 90(5), 588-599.
Thompson MB, Goldie KN. 1990 Conductance and structure of eggs of Adelie penguins Pygoscelis adeliae and its implications for incubation. Condor 92, 304–312.

---

## Round 0.2 · Minor Revisions

Dear authors,

I am glad to say that your manuscript has improved by the addition of the reviewers suggestions. I still have some concerns about the P distribution patterns that you described to have found, because if “the P concentration increases from the mammillary layer to prismatic layer in all analyzed eggshells”, how you explain that “the P concentration is homogeneous throughout the mammillary and prismatic layers but decreases “from the prismatic layer to the external layer” in Triprismatoolithus stephensi?. I mean that if the concentration increases from mammillary to prismatic layers, it cannot be homogeneous throughout them. So, please, revise that paragraph and offer a hypothetical explanation for such a variable P distribution, apparently not registered previously in birds. Moreover, in Triprismatoolithus stephensi you found P peaks “on the boundary between the eggshell and surrounding sediment” (which you interpret as the signal for the putative cuticle), and other two peaks “in the sediment and on the boundary between the external layer and prismatic layer”, probably suggesting that P is an important component of the dinosaur eggshell?.
I hope to see the revised version if this article very soon.

Regards,
Graciela Piñeiro

---

## Round 0.3 · accepted · Accept

Dear authors,

You have made good improvement of your manuscript, mainly on the aspects where both reviewers and editor have found several inconsistencies and weaknesses. Even though I still have some doubts about your interpretations, I saw that you were a little more cautious in this last version of the manuscript and leaved an open possibility in the sense that other hypotheses can arise from future new studies on the subject. However I must say that the topic of your manuscript is exciting and you provided a very complete examination of the data, particularly I would like to remark the inclusion of a taphonomic section that provide an approach to understand the preservation of so delicate structures as the putative cuticle (and possibly also the membrana testacea) in an environment where such taphonomic feature is uncommon to see. All these are enough reasons to consider that your article will be a very interesting (although somewhat controversial) contribution to the dinosaur reproductive behavior that surely will encourage future research on the presence of a cuticle-like layer in oviraptorid and alvarezsaurid dinosaur eggs. Thus, I will accept the publication of this paper in PeerJ. Congratulations!

With kind regards,
Graciela Piñeiro

#